# A SARS-CoV-2 Vaccine Designed for Manufacturability Results in Unexpected Potency and Non-Waning Humoral Response

**DOI:** 10.3390/vaccines11040832

**Published:** 2023-04-12

**Authors:** Elliot Campbell, Julie Dobkin, Louis J. Osorio, Afsal Kolloli, Santhamani Ramasamy, Ranjeet Kumar, Derek B. Sant’Angelo, Selvakumar Subbian, Lisa K. Denzin, Stephen Anderson

**Affiliations:** 1Center for Advanced Biotechnology and Medicine, Rutgers University, Piscataway, NJ 08854, USA; campbell@cabm.rutgers.edu; 2Macrotope, Inc., Princeton, NJ 08540, USA; 3Child Health Institute of New Jersey, Department of Pediatrics and Pharmacology, Rutgers Robert Wood Johnson Medical School, New Brunswick, NJ 08901, USA; juliedob@gsbs.rutgers.edu (J.D.); osoriolj@rwjms.rutgers.edu (L.J.O.);; 4Graduate School of Biomedical Sciences, Rutgers, The State University of New Jersey, New Brunswick, NJ 08901, USA; 5Public Health Research Institute (PHRI), New Jersey Medical School, Rutgers University, Newark, NJ 07103, USA; ak1482@njms.rutgers.edu (A.K.); santhuvet@gmail.com (S.R.); rk879@njms.rutgers.edu (R.K.); 6Department of Molecular Biology and Biochemistry, Rutgers University, Piscataway, NJ 08854, USA

**Keywords:** SARS-CoV-2, COVID-19, vaccine, durable immunity, emerging variants, protection

## Abstract

The rapid development of several highly efficacious SARS-CoV-2 vaccines was an unprecedented scientific achievement that saved millions of lives. However, now that SARS-CoV-2 is transitioning to the endemic stage, there exists an unmet need for new vaccines that provide durable immunity and protection against variants and can be more easily manufactured and distributed. Here, we describe a novel protein component vaccine candidate, MT-001, based on a fragment of the SARS-CoV-2 spike protein that encompasses the receptor binding domain (RBD). Mice and hamsters immunized with a prime-boost regimen of MT-001 demonstrated extremely high anti-spike IgG titers, and remarkably this humoral response did not appreciably wane for up to 12 months following vaccination. Further, virus neutralization titers, including titers against variants such as Delta and Omicron BA.1, remained high without the requirement for subsequent boosting. MT-001 was designed for manufacturability and ease of distribution, and we demonstrate that these attributes are not inconsistent with a highly immunogenic vaccine that confers durable and broad immunity to SARS-CoV-2 and its emerging variants. These properties suggest MT-001 could be a valuable new addition to the toolbox of SARS-CoV-2 vaccines and other interventions to prevent infection and curtail additional morbidity and mortality from the ongoing worldwide pandemic.

## 1. Introduction

More than three years have elapsed since the first cases of SARS-CoV-2 infection were reported in humans. Rapid transmission and continued evolution of the virus have led to a pandemic that persists to the present day. The first approved SARS-CoV-2 vaccines were remarkably effective against the ancestral strain, with multiple clinical trials demonstrating vaccine effectiveness at preventing severe disease of over 90% [1,2]. However, waning immunity and the emergence of new variants, many of which possess some degree of immune escape [3,4], has necessitated boosters and spurred the development of variant-specific and pan-coronavirus vaccines. Further, despite the availability of approved vaccines, accessibility has been problematic outside of the developed world, and hesitancy towards vaccines and new vaccine technologies have slowed vaccination rates everywhere. Finally, efficacious vaccines and strategies for members of the population who are immunocompromised remain a significant scientific and medical challenge.

Continued research and development of novel vaccines, adjuvants, and immunization strategies to combat these weaknesses remain a high priority [5,6]. The WHO Target Product Profiles for COVID-19 Vaccines was revised in April 2022 to reflect this need and described several desired characteristics for the next generation of vaccine candidates. Notable among these are the durability of protection, broader protection against emerging variants, and ease of manufacture and distribution. No current vaccine meets all of these criteria. Booster doses have been shown to enable protection against some emerging variants, but with the rapid waning of effectiveness and continued vaccine hesitancy [7,8] it is not clear whether current booster administration paradigms will comprise a sustainable strategy, even with variant-specific modifications to current vaccines [9,10]. 

Among the earliest vaccines approved in the US and EU were two mRNA vaccines from Pfizer/BioNTech (BNT162b2) and Moderna (mRNA-1273), and two viral-vectored vaccines from Janssen/J&J (Ad26.COV2.S) and Oxford/AstraZeneca (ADZ1222). The mRNA vaccines BNT162b2 and mRNA-1273 elicit extremely high antibody titers [11], but studies have shown that the humoral immunity fades relatively quickly [12], prompting many countries to recommend a third booster dose and, presently, even a fourth or fifth booster in some cases [13]. Unfortunately, even with multiple boosts, protection against SARS-CoV-2 variants remains modest [14]. Conversely, the viral-vectored vaccines Ad26.COV2.S and AZD1222 elicit lower initial antibody responses [15], but protection seems to be more durable as immunological readouts remain relatively constant over time [12,16]. Perhaps most unexpectedly, and in stark contrast to the waning antibody titers observed for the mRNA vaccines, both the magnitude and breadth of the humoral immune response appear to increase with time after vaccination with Ad26.COV2.S [17,18]. The mechanisms mediating this non-waning behavior are unclear, but it may be due, at least in part, to differences in the kinetics of antigen presentation. The mRNA vaccines have been shown to produce a large bolus of short-lived spike protein [19], whereas the viral-vectored vaccines may provide modest, yet sustained, antigen levels over a more extended period [20]. 

The choice of immunogen remains an open question as well. Whereas most approved vaccines use the full-length SARS-CoV-2 spike protein as an immunogen, a strong argument can be made for a vaccine based on a smaller fragment of the spike protein encompassing the receptor binding domain (RBD). RBD-based SARS-CoV-2 vaccines have been shown to elicit a higher fraction of neutralizing antibodies (nAbs) than vaccines based on the full-length spike protein, likely due to the entire immune response being directed toward the RBD [21,22]. A high neutralizing titer is desirable, as numerous studies have shown that nAb levels strongly correlate with protection [23,24,25]. A comprehensive review of the potential advantages of RBD-based vaccines has been presented [26]. Despite the many potential benefits, existing RBD vaccine candidates have often suffered from relatively poor expression and/or reduced immunogenicity. Previous efforts to design RBD constructs have, at times, attempted to trim the domain down to the “minimal expressible unit” containing the receptor binding motif (RBM), either by inspection or based upon homology to constructs used for other coronavirus RBDs [27,28,29,30,31,32,33]. These approaches often truncate a significant portion of the local “context” of protein structure surrounding the RBM, which might negatively impact protein folding and stability. Several such constructs have been designed with key glycosylation sites knocked out, disulfides removed, or stabilizing mutations made within the structure in order to rescue protein expression [27,28,30]. However, such changes may lead to an immunogen 3D structure that differs from the native conformation of the target viral protein antigen. This could potentially negatively impact antigenicity and thus the utility of the vaccine. 

Bearing these considerations in mind, we designed a novel protein component vaccine based on the RBD and RBD-adjacent sequences of the SARS-CoV-2 spike protein. By focusing the immune response on the region of the spike protein, where the bulk of the epitopes for neutralizing (including broadly neutralizing) antibodies reside [34,35], we aimed to enable high potency. We also sought to design a recombinant immunogen that would be stable, highly soluble, capable of expression at high levels, and amenable to streamlined purification protocols. We reasoned that this would endow the vaccine with relatively uncomplicated manufacturing and distribution requirements that would facilitate its adoption on a global scale. 

We show here that with a 2-dose prime-boost regimen in BALB/cJ mice, the resultant vaccine, MT-001, exhibited peak anti-spike IgG ELISA titers comparable to those reported in studies with mRNA vaccines from Pfizer/BioNTech (BNT162b2) and Moderna (mRNA-1273) at similar doses in the same animal model [36,37]. When adjuvanted with both aluminum hydroxide (Alhydrogel) and the TLR-9 CpG agonist ODN1826, the MT-001 vaccine in BALB/cJ mice showed a balanced Th1/Th2 response as well as peak anti-spike RBD IgG midpoint ELISA titers on the order of 10^6^ GMT. Syrian golden hamsters vaccinated with MT-001 adjuvanted with alum plus CpG exhibited undetectable levels of SARS-CoV-2 in lung tissue four days after intranasal challenge with SARS-CoV-2/US-WA1. We also observed that anti-spike IgG ELISA titers in sera from vaccinated mice were durable, with EC_50_s in the range of 10^5^–10^6^ up to 12 months post-vaccination. Furthermore, the results showed a meaningful breadth of the response, with significant neutralization titers against the Omicron BA.1 variant at this time point.

Combined, these attributes make MT-001 a compelling candidate for further research and development as a next-generation COVID-19 vaccine. MT-001 (or variant-updated versions thereof) could be particularly valuable as an annual booster to augment immunity in individuals with diverse histories of vaccination, SARS-CoV-2 infection, and/or predispositions resulting in an immunocompromised state.

## 2. Materials and Methods

### 2.1. Design and Expression of MT-001

The sequence of the ancestral SARS-CoV-2 Wu-1 strain spike protein (YP_009724390.1) was analyzed using publicly available bioinformatics tools for calculating structural, biophysical, and biochemical properties of potential constructs. Access to such tools can be found on the DisMeta server [https://montelionelab.chem.rpi.edu/ (accessed 19 September 2022)]. Shown in Appendix A is an example output from DisMeta for residues 300–600 of the spike protein. The results of these analyses were used to parse the sequence to yield a final expression construct designed to encompass the annotated receptor binding domain (residues 319–541), but with the construct N- and C-termini extended to include additional spike protein structural elements flanking the RBD domain that might promote proper domain folding and improved stability. The resulting RBD construct, MT-001, corresponded to residues 316–594 of spike fused to a C-terminal C-tag. The MT-001 construct was codon-optimized and expressed via a secretion vector in HEK293 cells (ATUM, Inc., Newark, CA, USA), and purified in a single affinity chromatography step using the CaptureSelect C-tagXL system (ThermoFisher Scientific, Waltham, MA, USA) [38,39]. The final purified yield was >160 mg from 1 L suspension culture. The purified protein was >96% monomeric with an apparent molecular weight of 39.4 kDa (calculated 31.6 kDa) by HPLC-SEC and had an apparent purity of >99% by capillary electrophoresis (Appendix A). Solubility was determined to be >10 mg/mL in PBS (137 mM NaCl, 2.7 mM KCl, 8 mM Na_2_HPO_4_, and 2 mM KH_2_PO_4_, pH 7.4). Aliquots were formulated in PBS with 10% glycerol as a cryoprotectant and stored at −80 °C until use.

### 2.2. Immunization of BALB/cJ Mice

Cohorts of 5–10 female, 8–10 week old BALB/cJ mice (The Jackson Laboratory, Bar Harbor, ME, USA) were immunized by injection into the gastrocnemius muscle with the indicated amount of MT-001 adjuvanted in 500 µg Alhydrogel^®^ (InvivoGen, San Diego, CA, USA) in a final volume of 50 µL. Mice were boosted 21 days later with an injection of the same MT-001/Alhydrogel dose. Where indicated, 20 µg of CpG-ODN1826 (InvivoGen, San Diego, CA, USA) was added to the MT-001/Alhydrogel mix immediately before immunization. Pre-immune sera were collected 3 days prior to the initial immunization, and immune sera were collected after immunizations, as indicated in each figure. 

### 2.3. RBD-Binding ELISA

RBD-specific IgG antibody levels were assessed using a novel sandwich ELISA (Appendix A). This assay was developed to maintain 3D conformational epitopes of RBD and prevent the loss of epitopes that may be denatured by direct adsorption of protein to plastic. Plates were coated with 1 μg/mL streptavidin (Sigma-Aldrich, St. Louis, MO, USA) diluted in PBS and incubated at 4 °C overnight. The next day, plates were washed three times with 0.1% TWEEN-20 in PBS (PBS-T), blocked with PBS-T containing 3% BSA for 1 h at room temperature, and incubated at 4 °C overnight with 1 μg/mL biotinylated-camelid α-C-Tag-specific antibody (ThermoFisher Scientific) in PBS-T. Plates were washed, incubated for 1 h at room temperature with 5 μg/mL MT-001 (containing the C-tag) in PBS-T, washed and incubated at 4 °C overnight with serially diluted mouse serum in blocking buffer. Antibody levels specific to Delta variant RBD were assessed in mouse sera by direct ELISA: Plates were coated with 1 μg/mL Delta variant RBD (Leinco Technologies, Inc., Fenton, MO, USA) or MT-001 diluted in PBS and incubated at 4 °C overnight. Plates were washed, blocked, and incubated with serially diluted mouse serum in blocking buffer at 4 °C overnight, as described. To quantify total IgG levels in the ELISAs, plates were washed and incubated for 1 h at room temperature with goat anti-mouse IgG horseradish peroxidase (HRP) (Jackson ImmunoResearch Labs, West Grove, PA, USA): IgG1 and IgG2a/b levels were quantified using goat anti-mouse IgG1 HRP (Southern Biotechnology Associates, Inc., Birmingham, AL, USA) and goat anti-mouse IgG2a HRP (Southern Biotechnology Associates, Inc., 1080-05) with anti-mouse IgG2b HRP (Southern Biotechnology Associates, Inc.), respectively. ELISAs with hamster sera utilized goat anti-Syrian hamster IgG HRP (Jackson ImmunoResearch). All HRP-conjugated secondary antibodies were diluted 1:5000 in PBS-T. Finally, plates were washed, and 1-Step Ultra TMB-ELISA substrate solution (ThermoFisher Scientific) was added to each well to detect HRP activity. Development was halted by adding 1M sulfuric acid and absorbance at 450 nm was assessed using a SpectraMax i3 microplate reader (Molecular Devices, San Jose, CA, USA). Background values were recorded from wells containing block solution only and subtracted from the raw OD_450_ values. For ELISAs with mouse serum, a standard curve was derived using a titrated anti-RBD mouse monoclonal antibody (Sino Biological, Wayne, PA, USA) included on each plate as a technical control to monitor plate-to-plate variability. Data were analyzed in GraphPad Prism (GraphPad Software, Boston, MA, USA) using a sigmoidal four-parameter logistic (4PL) fit, and ELISA half-maximal titers were defined as the reciprocal serum dilution that yielded 50% maximal absorbance. ELISAs were repeated at least three times for each mouse or hamster serum sample, and the data represent average half-maximal titers for each set of replicates. Independent confirmation of the precision and accuracy of our indirect “sandwich” RBD ELISA method was obtained by submitting a test panel of mouse sera for analysis by Nexelis (Laval, Quebec, CA, USA) using a clinically validated SARS-CoV-2 anti-spike IgG ELISA assay [https://nexelis.com/our-expertise/infectious-diseases/vaccine/sars-cov-2/ (accessed 10 August 2021)]. Replicate serum samples assayed by the anti-RBD IgG ELISA at Rutgers (above) and, in parallel, an optimized automated anti-spike IgG ELISA at Nexelis yielded highly concordant results with quantitatively similar titers (Appendix A).

### 2.4. Propagation of SARS-CoV-2

Vero E6 cells (ATCC, Manassas, VA, USA) were propagated in DMEM (Sigma-Aldrich) containing L-glutamine and 10% FBS (Sigma-Aldrich) to 80% confluency in multiple T75 flasks (Corning Inc., Corning, NY, USA), and harvested by gentle dissociation of the monolayer with Accutase Cell Detachment Solution following the instructions of the manufacturer (ThermoFisher Scientific). Pooled cells were washed twice in sterile PBS (pH 7.2) and checked for viability by the Trypan Blue (ThermoFisher Scientific) exclusion method. Cells were seeded into T75 flasks to ~80% confluency in DMEM containing 10% FBS, and after 18 h, the spent media was decanted, and the cells were washed with sterile PBS (pH 7.2). To determine the viral titer, the original stock vial of SARS-CoV-2/USA-WA1/2020 strain (BEI Resources, Manassas, VA, USA), obtained as lysate of infected cells, was diluted in DMEM containing 2% FBS and used for infection as we described previously (40). Briefly, about ~8 × 10^6^ Vero cells in a T75 flask were infected with 1 mL of virus suspension and incubated at 37 °C for 1 h, followed by replenishing cells with 10 mL of DMEM containing 2% FBS. The cell culture supernatant containing the virus was harvested at 72 h post-infection by centrifugation, followed by filtration using a 0.4-micron filter (Millipore-Sigma, Burlington, MA, USA). Aliquots of virus-containing media (inoculum) were stored at −80 °C until ready to use. The infectious virus particles in the inoculum were quantitated by plaque assay (see below).

### 2.5. Virus Inoculum Titration

Virus infectivity and inoculum titer were quantitated by plaque assay using Vero E6 cells. Briefly, 4 × 10^5^ Vero cells/well were seeded onto a six-well cell culture plate (Corning) in DMEM supplemented with L-glutamine and 10% FBS. At 18 h post-seeding, the cells were washed with sterile PBS (pH 7.2), and 400 µL of 10-fold dilutions of the virus inoculum, prepared in serum-free DMEM, was added to each well and incubated at 37 °C with gentle rocking of plates every 15 min for 1 h. Then, the virus inoculum was carefully removed, and the infected cells were overlayed with 4 mL/well of 1.6% agarose prepared in DMEM with 4% FBS. The plates were allowed to solidify at room temperature for ~15 min and transferred to a 37 °C incubator with 5% CO_2_. At 3 days post-infection, the plates were fixed with 10% buffered formalin (VWR, Radnor, PA, USA) for 30 min and washed with sterile PBS (pH 7.2). The agar plugs were gently removed, and the cells were stained with 0.2% crystal violet in 20% ethanol (VWR) for 10 min. The wells were washed with sterile water and dried, and the clear plaques were counted and presented as the number of plaque-forming units (PFU) of the virus per gram or ml of tissue or lysate.

### 2.6. Hamster Infection Studies

Five-to-six-week-old male golden Syrian hamsters (*Mesocricetus auratus*) were procured from Envigo (Indianapolis, IN, USA) and housed in animal biosafety level-2 containment (BSL2) for a week to acclimate. One group of hamsters (*n* = 8) was vaccinated with adjuvanted MT-001, and another group of hamsters (*n* = 6), injected with PBS plus Alhydrogel, served as the control. The MT-001 vaccine or PBS was mixed with Alhydrogel and incubated for 5 min with gentle rocking. Then, the MT-001/Alhydrogel and the PBS/Alhydrogel mixtures were supplemented with CpG-ODN1826 immediately prior to injection. Each hamster was injected intramuscularly, in the flank, with 50 µL of the respective RBD or control vaccines containing 10 µg of MT-001 (or an equal volume of PBS), 500 µg of Alhydrogel, and 100 µg of ODN1826. The hamsters were administered a second dose of MT-001 or PBS control 21 days after the primary dose. Blood from all animals was collected on the day of vaccination (day 0; pre-bleed) and at 14, 21, 28, 35, and 42 days post-vaccination. On day 42, post-primary vaccination, all animals were challenged with SARS-CoV-2/USA-WA1/2020 strain (BEI Resources) at 10^5^ PFU/hamster in 40 µL through intranasal instillation (20 µL/nostril) as we reported previously [40]. Hamsters were weighed every day following infection and euthanized on day 4 post-infection. Necropsy was performed, and blood and lungs were collected under aseptic conditions. 

### 2.7. Lung Viral Load Assessment

Lung homogenates were prepared using a 0.3 mg (~40% total lung weight) portion of lung tissues in a screw cap vial containing 1 mL of DMEM media and 0.3 mL (*w*/*v*) of 1 mm Zirconia/silica beads (MP Biomedicals, Irvine, CA, USA). Tissues were lysed by using a FastPrep homogenizer (MP Biomedicals). The homogenates were centrifuged, and the supernatant was filtered through a 0.45-micron filter (Millipore-Sigma), diluted in serum-free DMEM, and 400 µL was used to infect Vero E6 cell monolayers in the six-well plates for a virus plaque assay. 

### 2.8. Determination of Viral Load by Quantitative PCR

Total RNA was extracted from the lungs using TRIzol reagent (ThermoFisher Scientific) and purified by RNeasy mini columns (Qiagen Sciences Inc., Germantown, MD, USA). The eluted RNA was subjected to complementary DNA synthesis using a High-Capacity cDNA Reverse Transcription Kit as per the suggested protocol (ThermoFisher Scientific). Quantitative PCR (qPCR) was performed as described by Ramasamy et al. [40] using SARS-CoV-2 N gene-specific primers (SARS-CoV-2_N-F1: GTGATGCTGCTCTTGCTTTG and SARS-CoV-2_N-R1: GTGACAGTTTGGCCTTGTTG) (IDT, Coralville, IA, USA) and Power SYBR Green PCR MasterMix as per the manufacturer’s protocol (ThermoFisher Scientific). The SARS-CoV-2 N gene-specific primers were used to amplify a 97 bp product by conventional PCR and this was purified by the Qiagen gel extraction kit (Qiagen). The purified N gene PCR products were used in real-time PCR to prepare a standard curve. Viral copy numbers in the lung samples were determined from the standard curve.

### 2.9. Virus Neutralization Assay

The SARS-CoV-2 neutralization assay was performed using the standard protocol described by Ravichandran et al., 2020 [41]. Briefly, 100 TCID_50_ of SARS-CoV-2 isolate USA-WA1/2020 or Omicron variant (B.1.1.529) was added to a two-fold dilution series of serum samples in DMEM containing 10% fetal bovine serum. The serum-virus mixtures were incubated at 37 °C for 1 h. Meanwhile, a single-cell suspension of Vero E6 cells was prepared in DMEM containing 10% fetal bovine serum at 1.4 × 10^4^ cells in 20 µL/well in white 96-well flat-bottom Nunc MicroWell plates (ThermoFisher Scientific). Following incubation, 100 µL of the serum-virus mixture was added to each well. Additional wells omitting either the serum samples or the virus were included as controls. The plates were gently rocked for the uniform distribution of cells and then incubated for 72 h at 37 °C with 5% CO_2_. Plates were equilibrated to room temperature for 30 min, after which 50 µL of CellTitre Glo reagent (Promega, Madison, WI, USA) was added to each well, and the plates were gently rocked for 2 min and incubated at room temperature for 10 min. The luminescence from the wells was measured using Cytation 5 Cell Imaging Multi-Mode Reader (BioTek Instruments, Winooski, VT, USA). The luminescence from blank wells containing 120 µL DMEM with 10% fetal bovine serum and 50 µL CellTitre Glo reagent was recorded as baseline values. The 50% neutralization titer (NT_50_) was calculated using Graph Pad Prism from a sigmoidal four-parameter logistic (4PL) fit the luminescence data using the geometric means of the positive and negative controls to bind the top and bottom of the curve. 

### 2.10. Histopathology

The formalin-fixed hamster lung portions were embedded in paraffin and sectioned following standard protocol, as we reported previously [40]. The hematoxylin-eosin-stained lung sections were analyzed using the EVOS FL cell imaging system (ThermoFisher Scientific). The pulmonary inflammation was scored according to the severity as follows: 0—no cellular infiltration and intact alveoli, 1—mild cellular infiltration with one or two foci and intact alveoli, 2—prominent multifocal cellular infiltration with no visible alveoli, 3—significant cellular infiltration involving a larger area of the lung with no visible alveoli, and 4—highest cellular infiltration involving extensive area of the lung with no visible alveoli.

## 3. Results

### 3.1. Antigen Construct Design Impacts Both the Manufacturability and Immunogenicity of a Protein Component Vaccine

Upon the publication of the ancestral SARS-CoV-2 Wu-1 strain DNA sequence in early 2020 [42], we applied antigen expression construct design principles established in the lab, aiming to create a well-folded and soluble spike RBD antigen based on a fragment of the S1 subunit. The design of the construct is critical when parsing a multi-domain protein into smaller expressible subunits [43]. The lab previously provided over 1000 unique human antigens to the *NIH Common Fund Protein Capture Reagents Program* for renewable antibody generation [43,44]. Central to this effort was a bioinformatics toolbox, developed by the Northeast Structural Genomics Consortium, for parsing multi-domain proteins into subdomains that could be expressed recombinantly [45]. These tools, involving meta-analyses of protein amino acid sequences using various protein structure prediction methods, have been used successfully to design and optimize thousands of protein constructs for NMR and crystallization studies [46] as well as antigens for antibody discovery [43]. In all cases, domain boundaries and other sequence features were given special weight, so as not to truncate constructs within ordered regions required for proper folding or presentation of conformational epitopes [28,29,30]. 

We reasoned that an immunogen designed to preserve domain structure would enhance expression yields and promote optimal manufacturability. By combining bioinformatics predictions from DisMeta [45] with protein homology models and sequence alignments to known structures, we identified clear domain boundaries that separated the RBD region from the surrounding N-terminal and C-terminal regions of the spike protein. The resulting fragment, consisting of residues 316–594 of the full-length SARS-CoV-2 spike protein, encoded a 279 amino acid polypeptide with two complex subdomains containing non-contiguous N-terminal and C-terminal residues distal to the RBD ACE2 binding region (Figure 1; PDB IDs: 7BYR, 7KNE). In addition to the so-called CD1, RBM (receptor binding motif), and CD2 regions (Figure 1), this fragment also included the region immediately C-terminal to the RBD, previously termed C-terminal domain 1 (CTD1), and a portion of the so-called “N-terminal domain” of S1 in SARS-CoV [47]. A short, four-residue “C-tag” [-EPEA] was appended to the C-terminus of the fragment to facilitate efficient purification from cell culture [38], and the resulting construct was termed MT-001. No linkers or protease cleavage sites were included in order to minimize the number of non-native residues in the expressed protein. As the N- and C-termini of the construct are predicted to be located on the face of the protein opposite the RBM (Figure 1), it was thought to be unlikely that the short C-tag would sterically hinder desired antibody interactions. The C-tag also provided a convenient site-specific handle for immobilization when used in downstream assays (see ELISA in Methods). Finally, the immunogenicity of the C-tag has been investigated, and no significant anti-C-tag antibody responses have been observed [39]. Transient expression of MT-001 with a mammalian cell secretion vector (ATUM, Inc.) in HEK293 suspension culture resulted in high titers of MT-001, as described in Methods. The purified protein was nearly all monomeric, with an apparent molecular weight of 39.4 kDa, consistent with what would be expected for a glycosylated protein (calculated unglycosylated MW = 31.6 kDa) (Appendix A).

### 3.2. MT-001 Induces a Potent and Durable Anti-SARS-CoV-2 RBD Immune Response in BALB/cJ Mice

To explore the immunization dose-response characteristics and measure the durability of elicited antibody levels, an experiment was performed in which two cohorts of five 8- to 10-week-old female BALB/cJ mice were immunized with 1 μg, 3 μg, or 15 μg of MT-001. The MT-001 immunogen was formulated with 500 μg Alhydrogel (alum) and administered as two intramuscular (IM) injections at a 3-week interval (Figure 2A). Sera were collected at 5, 29, and 52 weeks following the primary immunization (Figure 2A). The highest RBD-specific IgG half-maximal geometric mean titers (GMTs) at each time point were observed with the 3 μg and 15 μg doses of MT-001 (EC_50_ > 10^5^, Figure 2B). MT-001 at the 3 μg dose exhibited half-maximal ELISA GMTs comparable in potency to reported 2-shot prime/boost immunization results with approved mRNA vaccines and protein component vaccines assayed in the same BALB/cJ mouse system [36,37,48,49]. Most notably, there was no significant waning of the MT-001-induced specific anti-RBD antibody levels in the animals between 5 weeks and 52 weeks post-immunization (Figure 2). This is in contrast to the mRNA- and most other protein component-based vaccines where protective antibody levels typically wane with a half-life of approximately two months [50,51].

### 3.3. Addition of a TLR-9 Agonist CPG ODN1826 to the MT-001 Vaccine Mixture Further Increases Antibody Titers and Promotes a More Balanced Immune Response

Since alum-based adjuvants such as Alhydrogel promote a type 2 inflammatory response [52], we next tested if the addition of a TLR-9 agonist, CpG ODN1826, would promote a more balanced immune response. Mice immunized with 3 μg MT-001 formulated with 500 μg Alhydrogel and 20 μg CpG ODN1826 exhibited significantly increased RBD-specific IgG binding titers compared to mice immunized with MT-001 and Alhydrogel alone by 5 weeks post-primary immunization (ELISA GMTs ≈ 2 × 10^6^ for mice where CpG ODN1826 was included vs. ≈ 3.5 × 10^5^ when omitted), and the enhanced response persisted through 47 weeks (Figure 3B). In addition to significantly higher levels of RBD-specific IgG1 antibodies (Figure 3C), these mice had robust RBD-specific IgG2a/b titers (Figure 3D). Thus, the average IgG1:IgG2a/b ratio in mice adjuvanted with both Alhydrogel and CpG ODN1826 was significantly increased (Figure 3E), indicative of a more balanced Th1/Th2 response, which may strengthen the protective efficacy of MT-001. 

### 3.4. MT-001 Protects Syrian Golden Hamsters in a SARS-CoV-2 Pulmonary Challenge Model

Syrian hamsters are an accepted in vivo model for human SARS-CoV-2 infection as they mimic many of the characteristics of human COVID-19 [53]. Therefore, we next tested the protective efficacy of MT-001 against SARS-CoV-2 infection in vivo. Hamsters were immunized with 10 µg MT-001 adjuvanted with 500 µg Alhydrogel and 100 µg CpG ODN1826 and boosted with the same dose 3 weeks later (Figure 4A). Control animals were only mock vaccinated with the adjuvant (Alhydrogel plus CpG ODN1826). The MT-001 vaccinated hamsters had robust RBD-specific IgG titers after six weeks post-primary immunization (EC_50_ ≈ 10^5^, Figure 4B). To determine if the antibody response was protective, the vaccinated and control hamster cohorts were challenged intranasally with 10^5^ PFU of SARS-CoV-2 (USA-WA1 strain) after six weeks post-primary immunization and monitored for 4 days before analysis. The level of infectious SARS-CoV-2 in lung homogenates from MT-001 vaccinated hamsters was undetectable by plaque assay even at the lowest dilution (1/10) of the sample used (Figure 4E). Therefore, the viral load per gram of lungs in these animals was calculated based on the lower limit of detection of the assay. In the control group of mock-vaccinated hamsters, infectious virus plaques were detected even at a 1:10^6^ dilution of lung homogenates. Thus, the viral load per gram of lung tissue was significantly lower (less than or equal to 10^3^ PFU/g) in MT-001 vaccinated hamsters than in mock-vaccinated hamsters (10^9^ PFU/g) (Figure 4C). Likewise, N gene copy numbers as determined by qPCR were on average 1000-fold lower in MT-001 vaccinated hamsters compared to hamsters that received adjuvant alone (Figure 4D). While weight loss and lung pathology are usually associated with SARS-CoV-2 infection in hamsters, at the viral dose used there were no significant differences between MT-001 vaccinated and mock-vaccinated animals with respect to these two parameters up to 4 days post-infection when the hamsters were sacrificed (Appendix A–C). However, a significant reduction in viral burden was observed in hamsters vaccinated with MT-001 compared to those that received adjuvant alone. Compared to the uninfected group, hamsters in both the MT-001 vaccinated and adjuvant-vaccinated groups showed a significantly higher degree of lung inflammation, marked with infiltration of immune cells into the interstitial space that resulted in the partial collapse of alveoli at 4 days post-infection (Appendix A). It should be noted that previous SARS-CoV-2 challenge studies have indicated that the lung pathology between vaccinated and mock-vaccinated hamsters does not begin to differ until four to six days post-challenge [54,55]. However, despite the difference in lung viral load, no striking differences were noticed in the lung disease pathology (Appendix A) or physiological aspects (body temperature and general locomotor skills) between MT-001 vaccinated and mock-vaccinated animals. Collectively, these studies show that vaccination with MT-001 protected hamsters from SARS-CoV-2 infection.

### 3.5. Immunization with MT-001 Produces a Broad Antibody Response Capable of Recognizing and Neutralizing Emergent Variants including Delta and Omicron

We then asked if the response in mice vaccinated with MT-001 resulted in antibodies that were reactive to emerging variant SARS-CoV-2 strains. We first compared serum anti-RBD IgG ELISA titers directed against the ancestral SARS-CoV-2 Wu-1 (“WT”) strain to IgG ELISA titers from the same sera with the SARS-CoV-2 Delta variant RBD. Sera from mice vaccinated with MT-001 adjuvanted with Alhydrogel + CpG ODN1826 showed less than a 4-fold decrease in titer with Delta RBD when compared to the titers obtained for WT (Wu-1/US-WA1) RBD (GMTs: 194,082 vs. 739,163, respectively; Figure 5A). Vaccination without the inclusion of the CpG ODN1826 co-adjuvant, however, resulted in a more than 20-fold decrease in the Ab binding titer to the Delta RBD as compared to WT RBD (ELISA GMTs: 10,237 vs. 223,688, respectively). This indicated that the magnitude and breadth of the cross-reactive antibody response to variants elicited by MT-001 were enhanced by the inclusion of CpG ODN1826.

We next tested if the enhanced recognition of the Delta variant RBD by sera from mice vaccinated with MT-001 adjuvanted with Alhydrogel + CpG ODN1826 correlated with an enhanced ability to neutralize the Omicron BA.1 SARS-CoV-2 variant. In a live virus in vitro neutralization assay, sera from mice immunized with MT-001 adjuvanted with Alhydrogel + CpG ODN1826 had an Omicron BA.1 virus neutralization titer (NT_50_) of 2092 at six months post-boost and 1456 at 11 months post-boost (Figure 5B). When CpG ODN1826 was not included, the NT_50_s were reduced to 171 at six months post-boost and 190 at 11 months post-boost. Strikingly, the Omicron BA.1 neutralization titers obtained with MT-001 adjuvanted with Alhydrogel + CpG ODN1826 were comparable to neutralizing titers reported in BALB/c mice immunized twice with a variant-matched vaccine, mRNA-1273.529 [56]. These data show that MT-001 was efficacious in generating significant nAb responses against emergent variants, despite being based on the ancestral SARS-CoV-2 sequence, and that these nAb responses were durable for at least 11 months.

## 4. Discussion

Designing an expression construct that incorporates a fragment of the SARS-CoV-2 spike protein, encompassing both the spike ACE2 receptor binding motif (RBM) as well as surrounding sequences that provide the local structural context, is not a straightforward task [30]. Ideally, the design should result in good expression yields of a relatively “well-behaved” (i.e., stable, well-folded, and soluble) gene product while maximizing antigen immunogenicity and preserving conserved regions that might serve as targets for broadly neutralizing antibodies. Our MT-001 RBD expression construct (Figure 1), which includes the spike protein RBM (residues 437–508, green) together with upstream and downstream regions (residues 316–436, red and magenta, and residues 509–594, cyan and blue), encodes a section of the spike protein with an extended polypeptide architecture that appears to be composed of three distinct domain-like regions (Figure 1). The term “domain-like” in this sense refers to compact, structurally contiguous subdomain regions of the protein that may have distinct structural and/or functional roles. For example, ACE2 receptor binding is carried out by the domain-like RBM [57]. The extended three-subdomain structure is tethered at its N- and C-termini by residues S316 and G594 in close proximity, forming the ends of a short antiparallel beta-sheet (Figure 1). At several points within some of the subdomains, residues that are relatively distant from the primary sequence form significant interactions in the secondary and tertiary structure. For example, in the central subdomain domain (Figure 1, magenta), which consists of a twisted five-stranded antiparallel beta-sheet composed mostly of CD1 amino acids, the center beta strand (Figure 1, cyan) comes from the CD2 region C-terminal to the RBM. Additionally, the CTD1 subdomain (the region from 528 to 594, Figure 1, blue) in our construct forms a well-defined structure, stabilized by the 538–590 disulfide bond (Figure 1B, red arrow), and packs against the N-terminal region (“NT”, 316–332) of the MT-001 spike fragment (Figure 1, red). 

These interactions likely play an important role in the proper folding of the RBD; shorter constructs, involving truncations that result in the loss of these interactions, might partially destabilize the native structure, and could even introduce non-native conformational epitopes. This domain architecture suggests that the spike S1 region represented by our RBD construct, spanning amino acids 316–594, may have evolved via two consecutive nested domain insertion events [58,59]. The resultant 316–594 region of the spike protein may have then undergone selection as a coherent structural and functional unit involved in the conformational transition between the “RBD-down” and “RBD-up” states of the prefusion spike trimer [60]. Thus, for an RBD-centric immunogen in a SARS-CoV-2 vaccine, the MT-001 construct is arguably close to the optimal choice. Furthermore, structural analysis of the full-length spike protein shows that CTD1 may act as a relay between the RBD and the fusion-peptide proximal region (FPPR) domains to trigger fusion upon receptor binding [61]. The proposed relay function of CTD1 suggests that some antibodies targeting this region might interfere with viral entry and thus have SARS-CoV-2 neutralization activity. Since this region is relatively conserved among sarbecoviruses and contains few mutations found in SARS-CoV-2 variants of concern (VOCs), it may be able to elicit broadly neutralizing antibodies (bnAbs) ([62,63]; Figure 1C).

The properties that make an antigen well-suited for expression and purification may also translate into improved immunogenicity in the context of vaccines. By providing an “extended” RBD-containing SARS-CoV-2 spike protein construct that is stable and well-folded without requiring non-native modifications to the sequence, the immune response can be focused on a critical region of the spike protein containing many neutralizing epitopes [22,63,64]. This strategy may, in addition, minimize decoy or immunodominant epitopes, steric hindrance, or possibly immune suppressive components of the full-length protein. Outbreaks of SARS-CoV and MERS-CoV earlier this century, combined with the periodic emergence of new SARS-CoV-2 variants of concern and the constant threat of future coronavirus pandemics, motivate the development of a broadly protective pan-coronavirus vaccine. Within the spike protein, the RBM shares low sequence homology across the coronavirus family, which is expected given the numerous hosts and range of cellular receptors targeted. However, some regions flanking the RBM are relatively highly conserved, especially within the CTD1 subdomain (Figure 1 and Appendix A). For an S1 sub-fragment RBD-based vaccine [26], the inclusion of the CTD1 subdomain should present additional conserved B-cell and T-cell epitopes which may provide broader pan-variant and pan-coronavirus responses compared to RBD constructs where this region has been truncated. Indeed, a recent study involving a hierarchical Bayesian regression model trained on more than 6 million SARS-CoV-2 genome sequences predicted that even for *future*, yet-to-emerge variants of concern (VOCs), mutations in the CTD1 subdomain were likely to be relatively rare due to their negative contributions to overall viral fitness [65]. 

The fusion of purification tags and other non-native sequences must also be considered when designing a vaccine construct. MT-001 employs the C-tag, a short four-residue (-Glu-Pro-Glu-Ala) tag incorporated at the C-terminus of the construct, to enable efficient purification and site-specific immobilization for use in downstream assays [66]. This provides several advantages over other commonly used purification tags. Due to its size, the C-tag would be expected to have minimal effect on protein expression and solubility, and tag cleavage may not be required. Studies have found the C-tag itself to be minimally immunogenic, and it has been used successfully in GMP processes for vaccine manufacturing [39]. Purification step-yields are high after only a single tag-specific affinity chromatography step and, unlike RBD constructs incorporating fusions to non-viral scaffolding moieties [30,32], nearly 100% of the expressed protein consists of the target SARS-CoV-2 antigen (Appendix A). The C-tag allows for indirect solid phase immobilization of the antigen (e.g., in microtiter plate wells) and, with the tag being located on the opposite side of the antigen from the receptor binding motif, allows for an unimpeded display of the native 3D antigen structure and efficient capture of antibodies recognizing discontinuous conformational B-cell epitopes of the RBD. 

Our animal experiments have shown that immunization with MT-001 markedly enhanced the production of IgG antibodies specific to SARS-CoV-2 spike proteins, with levels comparable to the most effective vaccines characterized in the literature to date [36,37,48]. In mice, following a two-dose immunization with as little as 1 µg MT-001 adjuvanted with Alhydrogel, high anti-RBD (Figure 2) and anti-spike IgG (Appendix A) titers were observed. These were associated with the increased production of neutralizing antibodies against both pseudovirus (unpublished results) and infectious virus (Figure 5). Moreover, as demonstrated in two independent experiments, these immune responses were remarkably durable, with minimal waning in antibody titers observed between 5 weeks and one-year post-primary immunization (Figure 2 and Figure 3). This is in stark contrast to the widely used mRNA vaccines, where antibody titers decay significantly after 6 months [67,68]. Considering the lack of durability observed for most COVID-19 vaccines to date, results from our long-term in vivo studies further differentiate MT-001 from other immunization strategies directed against SARS-CoV-2 (See Figure 1 of [69]). Durable immunity conferred by vaccines has been attributed to the generation of long-lived plasma cells (LLPCs) residing in bone marrow, which in some cases can express and secrete protective, pathogen-specific antibodies for decades [70,71]. It is possible that MT-001 is unusually capable, especially compared to other SARS-CoV-2 vaccines [71], in eliciting high levels of spike protein-specific LLPCs. 

In the hamster model of pulmonary SARS-CoV-2 infection, vaccination with MT-001 protected the animals by significantly reducing the lung viral burden. However, the body weight loss and pulmonary pathology of SARS-CoV-2 infection were comparable between adjuvant-only vaccinated and MT-001 vaccinated hamsters. This observation indicates that disease pathology is not directly proportional to the lung viral load and that vaccinations with an immunogenic adjuvant may not have direct effects in reducing pulmonary pathology. Nonetheless, our observation is consistent with other hamster SARS-CoV-2 challenge studies in the literature, where the lung pathology between vaccinated and mock-vaccinated hamsters does not begin to differ until four to six days post-challenge [54,55]. Further studies will be required to address the mechanistic basis of the immune response elicited by MT-001 at the cellular level.

Alum-based adjuvants such as Alhydrogel are known to elicit a type 2 inflammatory response [72], which may not be ideal for inducing protective immunity against pathogens [73]. Previously, CpG-containing oligonucleotides have been shown to induce a type 1 response by acting as Toll-like receptor 9 (TLR9) agonists, providing a more balanced Th1/Th2 response when used in conjunction with alum adjuvants [28,52,74]. Mice immunized with a dose of 3 µg MT-001 adjuvanted with Alhydrogel and the TLR9 agonist CpG ODN1826 exhibited significantly higher anti-RBD IgG titers at 5 weeks post-primary immunization compared to mice immunized with 3 µg MT-001 adjuvanted with Alhydrogel alone. This increased titer appeared to be primarily due to a two-order-of-magnitude increase in the anti-RBD IgG2a/b titers in CpG adjuvanted mice, which resulted in a more balanced IgG2a/b to IgG1 ratio. This is consistent with data reported for another RBD-based protein vaccine utilizing a different CpG oligonucleotide and alum as adjuvants [74]. The significantly increased IgG2a/b titer associated with the CpG adjuvant persisted for at least 29 weeks post-primary immunization and was correlated with increased neutralization titers against both the Wu-1 strain and variants of SARS-CoV-2 (Figure 3 and Figure 5; additionally, see below). 

CpG ODN1826, when used as a co-adjuvant with Alhydrogel, has previously been shown to enhance peak immunogenicity in mice and hamsters with RBD-based SARS-CoV-2 vaccines [74]. We have shown here that, in addition to enhancing the potency and Th1/Th2 balance of the immune response elicited by MT-001 (Figure 3), the inclusion of CpG ODN1826 also enhanced the antibody response to emerging variants (Figure 5). This could have been due to a direct enhancement of the breadth of the response (see below) or simply due to a mass action effect where the levels of pre-existing anti-variant antibodies were elevated above a threshold concentration in the serum where they were rendered detectable by the assays used. Further work will be required to determine the details underlying CpG augmentation of the immune response in this system. In addition, unlike the waning of immunity against variants seen with spike-based mRNA vaccines, a two-dose regimen of MT-001 elicits a diverse and protective antibody response that persists for at least eleven months. In one recently described experiment, where BALB/c mice were pre-immunized with two doses of the mRNA vaccine BNT162b2 and boosted on day 104 with the same vaccine, the *peak* post-boost neutralization titer against Omicron BA.1 was reported to be 2075 GMT (BioNTech Innovation Series Presentation, 29 June 2022). Here, we report a similar Omicron BA.1 virus neutralization titer (GMT 2092) in MT-001-immunized BALB/c mice at six months post-immunization without an additional booster dose (Figure 5). In the absence of an additional booster, mRNA-vaccinated BALB/c mice typically show virtually no detectable variant neutralization titers at a comparable time interval post-immunization [55]. 

The increased breadth of the immune response observed when the MT-001 immunogen is adjuvanted with both alum and CpG ODN1826 comports with data showing that TLR-9 agonists activate the innate immune system by signaling through IRF7 while also directly stimulating B cells and dendritic cells [73]. This is consistent with the view that adjuvants such as TLR agonists (perhaps necessarily in the presence of a co-adjuvant such as alum) promote B-cell maturation in germinal centers, leading to higher affinity and broader antibody repertoires [73]. It has further been suggested that imprinting by innate signals during vaccination, dependent on the type and structure of the immunogen, the adjuvant(s), and the mode of delivery, among other variables, may drive the durability of the immune response by promoting the creation of long-lived plasma cells in bone marrow [69]. In translational studies aimed at developing a human vaccine, caution must be exercised when interpreting murine results with TLR-9 agonists. Compared to mice, humans and other primates express TLR-9 in a more limited subset of immune cell types, chiefly plasmacytoid dendritic cells and B cells [75]. However, it is reassuring that, for at least one other RBD-based SARS-CoV-2 vaccine (“RBD-I53-50”), a careful comparison of results with alum plus a TLR-9 agonist (CpG-1018) in both mice (strain C57BL/6) and NHPs (rhesus macaques) has been published [76,77,78]. It is noteworthy that for several key immunological metrics, including peak neutralizing antibody titers against the parent SARS-CoV-2 strain, neutralizing antibody titers against variants, CD4 T-cell responses, Th1 cytokine responses, and protection in a virus challenge assay, comparable responses were observed in both mice and nonhuman primates for the RBD-I53-50 vaccine co-adjuvanted with alum and CpG-1018. This concordance is encouraging and suggests that the results presented here for MT-001 will have predictive value in translational preclinical and clinical studies.

Regarding the translational relevance of the preclinical animal data presented here to future expectations for a human vaccine, particularly with respect to the durability of the immune response, attention should be paid to recent Phase 2 clinical data presented for the Corbevax vaccine [79]. Corbevax, like MT-001, incorporates an RBD-based immunogen, although the construct used to express the antigen for Corbevax, compared to the MT-001 design, is truncated at both the N- and C-termini (332–549) and modified to remove an unpaired cysteine (C538A) [27,28]. Corbevax is also produced in yeast cells rather than in animal cells as is MT-001. However, like MT-001, Corbevax is co-adjuvanted with alum and a CpG TLR-9 agonist (CpG ODN1826 in mice; CpG1018 in humans). In BALB/c mouse studies, Corbevax, when adjuvanted with alum alone, exhibits only modest IgG titers and pseudovirus neutralization titers [80]; hence, clinical studies with this vaccine have focused exclusively on formulations incorporating both the alum and the CpG adjuvants. Recently published Phase 2 studies of Corbevax have shown that it, like MT-001, exhibits remarkable durability up to 12 months post-vaccination [79]. Another notable SARS-CoV-2 vaccine for comparison purposes is the SCB-2019 vaccine developed by Clover Biopharmaceuticals. The SCB-2019 immunogen is the full-length spike protein ectodomain (based on residues 1–1211 of the ancestral Wuhan-HU-1 strain), trimerized via a proprietary C-terminal tag derived from human collagen [49]. Like Corbevax, SCB-2019 is adjuvanted with alum and CpG1018. When used to immunize female BALB/c mice at a 3 µg dose with a simple prime/boost regimen three weeks apart, SCB-2019 exhibited excellent persistence of the antibody broad neutralization titers after 140 days ([81], Figure 4C). However, compared to durable MT-001 live virus neutralization titers after 6 months of approximately 2000 GMT against Omicron BA.1.1.529 (Figure 5B, this work), the SCB-2019-immunized mice, without a third dose, exhibited Omicron BA.1.1.529 pseudovirus neutralization titers of <100 GMT) at all the later time points (Table 2 [“No 3rd dose boost Control”] and Figure 4C in reference [81]). We have also successfully expressed, in good yield and at high purity, additional variants using the same basic expression construct design and protocols used for MT-001. For example, a recombinant antigen containing the 17 point mutations found in Omicron BA.4/5 spike protein in the region corresponding to residues 316–594 of the parental strain was expressed and purified (unpublished results). This demonstrates that it should be feasible to re-design and produce updated annual booster vaccines with our system, creating new immunogens as needed that reflect the sequence information concerning recently emerged SARS-CoV-2 variants.

Taken together, the above results strongly suggest that the vaccine durability results presented here for MT-001 in mice will translate to humans. Moreover, our results show, at least for the MT-001 construct studied, that the aluminum hydroxide adjuvant alone, without the CpG TLR-9 agonist co-adjuvant, is sufficient to endow the vaccine humoral immune response with the property of high durability (Figure 2 and Figure 3). Interestingly, the Omicron neutralizing antibody titers elicited by MT-001 are significantly higher (>20-fold) than those elicited by SCB-2019, even though the spike fragment sequence of MT-001 is entirely contained within the sequence of the SCB-2019 spike ectodomain sequence. The higher anti-Omicron titers observed in mice for MT-001 vs. SCB-2019 might be due to differences in the respective neutralization protocols (e.g., live virus assays for MT-001 vs. pseudovirus assays for SCB-2019). Alternatively, MT-001 may display to the immune system cryptic B-cell epitopes that, when buried in the 3D structure of the full-length trimeric spike ectodomain holoprotein, are effectively unavailable for neutralizing antibody elicitation. 

Immunization with the RBD-based MT-001 construct focuses the immune response on the RBD domain, which has been demonstrated to elicit a significantly higher proportion of neutralizing antibodies compared to immunization with the full-length spike protein [26]. This distinction may be especially important in the context of boosting immunity with a variant-matched vaccine after prior vaccination or infection, as mutations arising in the RBD are often associated with immune escape. A variant-matched vaccine based on the full-length spike possesses a high degree of similarity to ancestral strains and existing vaccines. Boosting with such a vaccine has been shown to drive an immune response to conserved regions among the variant and ancestral strains previously imprinted by vaccination or infection, leading to only modest anti-variant antibody titers [82]. Conversely, a variant-matched vaccine based on MT-001 would not contain many of these shared ancestral spike epitopes, and the resulting response to the variant RBD region would be expected to induce a much stronger variant-specific neutralizing response. Combined with the durable immunity shown here, a variant-matched vaccine based on the MT-001 construct may be ideal for use as an annual booster designed to provide continuing protection against future SARS-CoV-2 infections.

## 5. Conclusions

MT-001 was designed from inception for improved manufacturability using construct design techniques refined during the operation of a high-throughput human protein production pipeline [83,84]. High-yield streamlined GMP manufacturing using standard protocols and existing infrastructure widely available in the biotech and pharmaceutical industry (e.g., 2000 L bioreactors, production-scale protein purification systems, know-how, and associated ancillary equipment) should facilitate large-scale, cost-effective production of MT-001. The ability to neutralize emerging variants, combined with MT-001′s potent and durable immunogenicity, its favorable biophysical properties, and reduced logistical requirements for widespread distribution, make it an attractive candidate for further development on a global scale. Since the virus emerged in late 2019 the SARS-CoV-2 pandemic has enveloped the entire world and, as the virus continues to evolve and new variants emerge, medical countermeasures are still playing catch-up. Vaccines such as MT-001 could be in the vanguard of a future toolkit of impactful new vaccines and therapies that offer the promise of a globally coherent solution.

## Figures and Tables

**Figure 1 vaccines-11-00832-f001:**
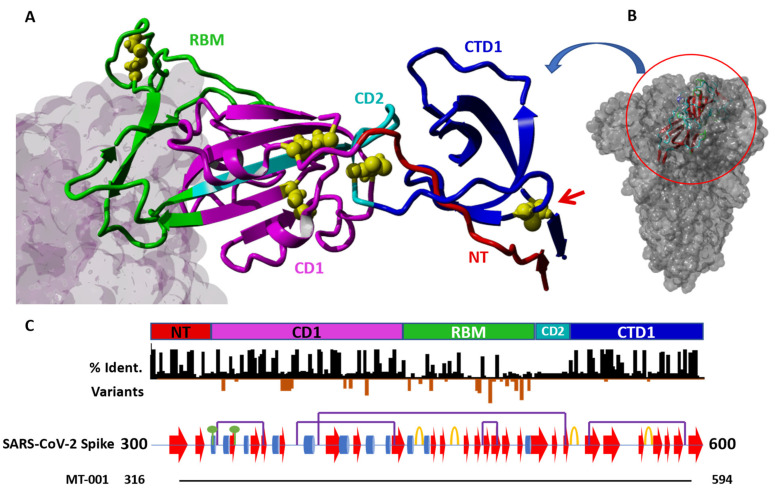
Detail of the SARS-CoV-2 spike protein in the region 300–600 and MT-001 construct design. (**A**) Structure of MT-001 construct derived from PDB IDs 7BYR and 7KNE. The RBD construct is color-coded by annotated blocks of amino acid sequence (“regions”, see panel C and [45]). Cysteines are shown as yellow balls. The cell surface target of the RBM, ACE2 (from 7KNE), is shown as a gray molecular surface (left). (**B**) The MT-001 construct (ribbon) shown in the context of the full-length spike trimer (space-filling model). (**C**) Schematic of the regions shown in (A). Top: Color-coded region key for the MT-001 construct in (**A**). NT: N-terminal region (residues 316–332, red); CD1: “core domain 1” region (333–436, magenta); RBM: receptor binding motif (437–508, green); CD2: “core domain 2” region (509–527, cyan); CTD1: “C-terminal domain 1” region (528–594, blue). The 538–590 disulfide bond that stabilizes CTD1 is indicated by a red arrow. Middle: Black bars—Sequence identity per residue between SARS-CoV-2 spike and representative members of the coronavirus superfamily, demonstrating highly conserved regions N- and C-terminal to the RBM (Appendix A). Orange bars—Sites of and frequency of mutations in characterized SARS-CoV-2 variants [3]. Lower: Schematic showing the secondary structure and post-translational modifications in the region from residues 300–600 in the SARS-CoV-2 spike protein. Alpha helices are shown as blue cylinders, beta sheets as red arrows, and turns as orange loops. Disulfide bonds are denoted with purple bridges, and N-linked glycosylation sites are denoted with green circles. Bottom: Alignment of the MT-001 construct with the visualized region.

**Figure 2 vaccines-11-00832-f002:**
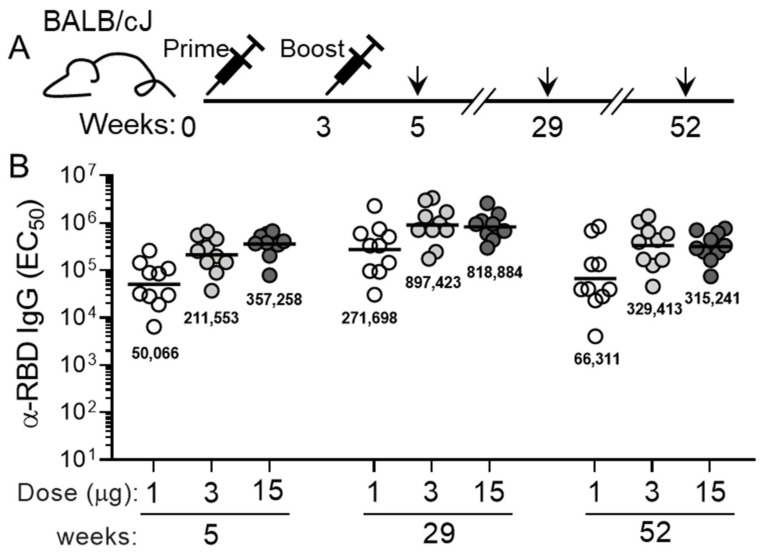
Dose-response and durability of anti-RBD serum IgG levels in mice vaccinated intramuscularly with Alhydrogel-adjuvanted MT-001. (**A**) Immunization and bleeding schedule. Prime and secondary immunizations of the animals were at weeks 0 and 3, respectively; bleeds were performed on week 5, week 29, and week 52 to provide sera for analyses. Primary and secondary immunizations were with the same amount of MT-001 antigen per animal—1 µg, 3 µg, or 15 µg—for each group of 10 mice. (**B**) Midpoint (EC_50_) geometric mean anti-RBD IgG ELISA titers for each dosage group at each time point. GMTs are indicated numerically below each cluster of points.

**Figure 3 vaccines-11-00832-f003:**
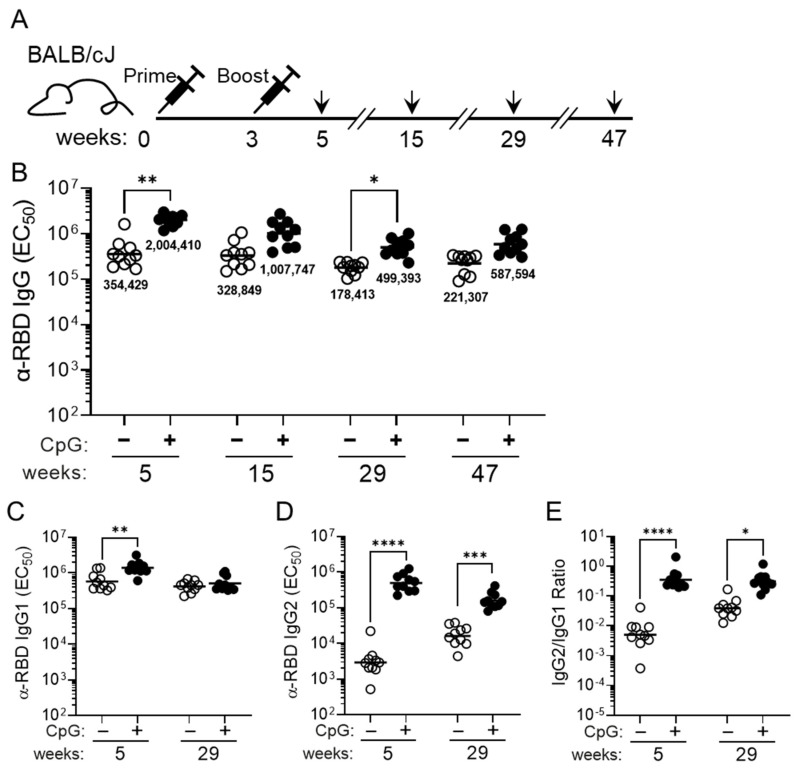
Immunization of mice with 3 µg of MT-001 antigen in Alhydrogel with or without the TLR-9 agonist co-adjuvant, CpG ODN1826. (**A**) Schematic illustrating the MT-001 prime-boost regimen and bleeding schedule for 8–10-week-old female BALB/cJ mice (*n* = 10). (**B**) RBD-specific IgG binding titers were assessed in mice immunized with 3 μg MT-001 adjuvanted with 500 μg Alhydrogel only (−, open circles) or with 500 μg Alhydrogel plus 20 μg CpG ODN1826 (+, closed circles). Binding antibody responses are displayed at 5, 15, 29, and 47 weeks post-primary immunization. The balanced Th1/Th2 response resulting from the addition of CpG ODN1826 is evidenced by increased RBD-specific IgG1 (**C**) and IgG2 (**D**) antibody titers. This corresponded to an increased ratio of IgG2 to IgG1 antibody levels in CpG ODN1826-adjuvanted animals (**E**). Each circle (open or solid) represents the half-maximal titer for each serum sample averaged across at least three independent ELISAs. Horizontal bars indicate geometric mean titers per dose. *p* values reflect unpaired *t* tests between groups (* *p* < 0.05; ** *p* < 0.01; *** *p* < 0.001; **** *p* < 0.0001).

**Figure 4 vaccines-11-00832-f004:**
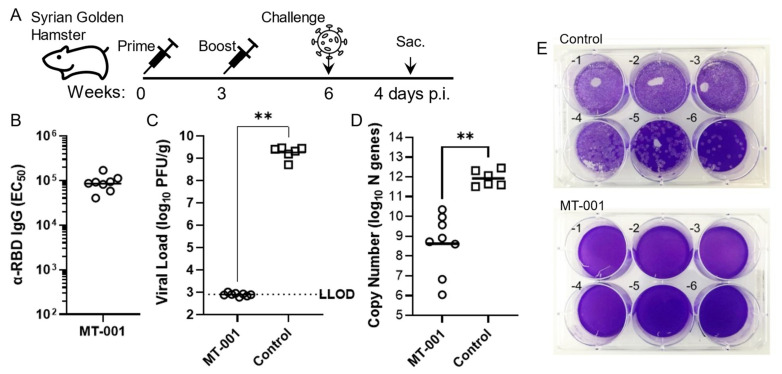
SARS-CoV-2 challenge of hamsters vaccinated with MT-001 adjuvanted with Alhydrogel and CpG ODN1826. (**A**) MT-001 prime-boost regimen and SARS-CoV-2 challenge schedule for Syrian golden hamsters. (**B**) Midpoint hamster RBD-specific IgG ELISA GMTs. (**C**) Lung viral load in hamsters vaccinated with 10 µg of MT-001 co-adjuvanted with 500 µg Alhydrogel + 100 µg CpG ODN1826, or mock-vaccinated with adjuvants alone, and infected with 10^5^ PFU of SARS-CoV-2 six weeks after the primary immunization (top). Individual data points and mean +/− S.D for MT-001 with adjuvants (open circles; *n* = 8) or adjuvants alone (open squares; *n* = 6) is shown. (**D**) Viral RNA copy numbers in the lungs of hamsters vaccinated with MT-001 or adjuvants alone four days after intranasal infection with SARS-CoV-2. Data were analyzed by non-parametric Mann–Whitney test (** *p* < 0.01). (**E**). Representative plates from lung viral load assessment. Lung homogenates from adjuvant-only (control) or MT-001-immunized (MT-001) hamsters were prepared 4 days post-infection and used to infect Vero E6 cells. No plaques are observed with the lung homogenates from MT-001 immunized hamsters even at a 1:10 dilution, while plaques are visible from the lungs of control animals even at a 1:1,000,000 dilution.

**Figure 5 vaccines-11-00832-f005:**
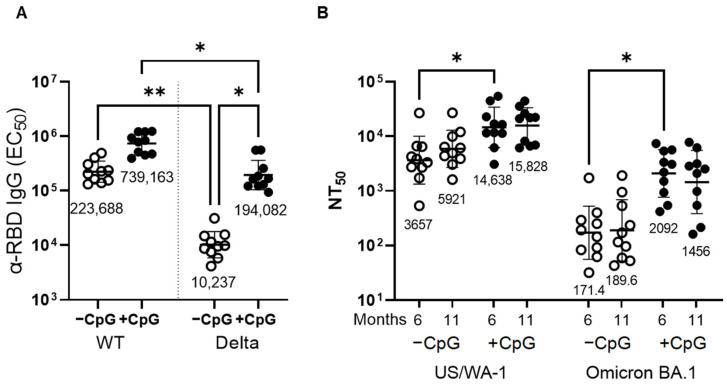
SARS-CoV-2 variant neutralization. BALB/cJ mice (*n* = 10 per group) were immunized twice at a three-week interval with 3 µg MT-001 and 500 µg Alhydrogel, with or without 20 µg ODN1826 (CpG), as indicated. Mice were bled at 29 and 47 weeks post-primary immunization (see Figure 3A), and sera were assayed for antibody binding and neutralization. (**A**) Anti-RBD midpoint ELISA titers at 29 weeks post-primary immunization were determined using the Wu-1 RBD (WT) or the Delta variant RBD (Delta) as a target. (**B**) Mouse serum-virus neutralization titers at approximately six months (29 weeks) and eleven months (47 weeks) post-primary immunization with MT-001 + Alhydrogel without (−CpG) or with (+CpG) ODN1826 were determined using SARS-CoV-2 USA-WA1/2020 (US/WA-1) or SARS-CoV-2 isolate hCoV-19/USA/MD-HP20874/2021 (Omicron BA.1) as described in Methods. Geometric mean titers are as indicated. Asterisks indicate statistical significance as determined by a two-tailed Kruskal–Wallis test with subsequent Dunn’s multiple-comparisons test (* *p* < 0.05; ** *p* < 0.01).

## Data Availability

The datasets used and/or analyzed during the current study are available from the corresponding authors upon reasonable request.

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
