# Peer review of "A SARS-CoV-2 Vaccine Designed for Manufacturability Results in Unexpected Potency and Non-Waning Humoral Response"

_vaccines, 2023, doi:10.3390/vaccines11040832_

Round 1

Reviewer 1 Report

In the present work, the manuscript's authors expressed a fragment of the spike gene expressed in the eukaryotic expression system and tested the efficacy of this protein in BALB/cJ 368 mice and Syrian golden hamsters. In challenge studies, the hamster showed protection post-inoculation. My concerns are as follows 

  1. How will you justify taking sequence YP_009724390? Since there are thousands of circulating Covid strains, considering a consensus sequence for vaccine development would be an intelligent choice. Since the outbreak, many VOCs have appeared, including alpha, beta, gamma, delta, and omicron, which outgrew the ancestral sars CoV 2. Should authors explain the rationale for using such an old sequence for vaccine development?
  2. I would like to see the expression profile of a protein expressing more than 160 mg/L of suspension culture. Authors may include the electrophoresis and western blot results in supplementary files.
  3. The authors did not mention the number of immunized animals. They wrote “Cohorts of 5-10 female”. It should be actual numbers.
  4. In the methods section how the authors decided the number of viruses in the stock from where they diluted the virus and inoculated
  5. Histopathological aspects are not very clear in the result and discussion section.

Author Response

Response to Comments from Reviewer-1:

General comment: In the present work, the manuscript's authors expressed a fragment of the spike gene expressed in the eukaryotic expression system and tested the efficacy of this protein in BALB/cJ 368 mice and Syrian golden hamsters. In challenge studies, the hamster showed protection post-inoculation.

Response: We thank reviewer 1 for the thoughtful review and comments on our manuscript. We will address the reviewer’s concerns point by point:

Comment-1: How will you justify taking sequence YP_009724390? Since there are thousands of circulating Covid strains, considering a consensus sequence for vaccine development would be an intelligent choice. Since the outbreak, many VOCs have appeared, including alpha, beta, gamma, delta, and omicron, which outgrew the ancestral sars CoV 2. Should authors explain the rationale for using such an old sequence for vaccine development?

Response-1: This vaccine development project began as soon as the first SARS-CoV-2 sequence was published, thus no variant sequences were available at the time. We share the reviewer's concern with vaccine effectiveness against emerging variants and were encouraged to see our vaccine was able to provide broad and durable protection against VOCs, despite being based upon the ancestral sequence (see Delta and Omicron BA.1 comparison data in Figure 5). The titers observed also compare favorably to the titers reported for variant-matched versions of the mRNA vaccines, again despite being based upon the ancestral sequence and assayed at later time points post-immunization. We do, however, believe that a variant-matched vaccine based on the construct presented here may be able to provide better protection than has been reported thus far for the variant-matched vaccines based on a full-length spike, as mentioned in the Discussion.

Comment-2: I would like to see the expression profile of a protein expressing more than 160 mg/L of suspension culture. Authors may include the electrophoresis and western blot results in supplementary files.

Response-2: We agree the expression levels are quite remarkable and present the expression and purification data in Figure S2. As the paper title reflects, we designed MT-001 for manufacturability (including high-level expression and efficient purification) and were excited to discover it was also capable of priming a potent and durable immune response.

Comment-3: The authors did not mention the number of immunized animals. They wrote “Cohorts of 5-10 female”. It should be actual numbers.

Response-3: The cohort sizes in the methods section were given as ranges, as they correspond to multiple experiments. The figure legends include the exact number of animals used in each experiment.

Comment-4: In the methods section how the authors decided the number of viruses in the stock from where they diluted the virus and inoculated.

Response-4: The virus titers were determined by PFU assay as described in the methods section. As suggested by the reviewer, we have edited the manuscript to provide more details of this assay in the methods section.

Comment-5: Histopathological aspects are not very clear in the result and discussion section.

Response-5: As suggested by the reviewer, we have revised the results and discussion sections pertaining to the histopathological analysis in the manuscript to provide more detail. We also revised Figure S3 and revised the legend of this figure to better present and explain the data.

Reviewer 2 Report

The article is well written and easy to follow. The methods presented to verify the antibody response seems adequate to show the efficacy of the new construct. The construct for spike RBM , the use of adjuvant and ODN seems to have given a good result in mice, but as mentioned in the discussion part , the effectiveness in humans are to be further evaluated.

The results given in supplementary information were not explained well in the captions of the figures. The subcategory of each figure were not mentioned in the captions, which could be improved upon for facilitating better grasp of the results.

Author Response

Response to Comments from Reviewer 2:
General comment: The article is well written and easy to follow. The methods presented to verify the antibody response seems adequate to show the efficacy of the new construct. The construct for spike RBM, the use of adjuvant and ODN seems to have given a good result in mice, but as mentioned in the discussion part , the effectiveness in humans are to be further evaluated.

Response: We are pleased reviewer 2 found the manuscript to be well-written and easy to follow, and that they found the results compelling.

Comment-1: The results given in supplementary information were not explained well in the captions of the figures. The subcategory of each figure was not mentioned in the captions, which could be improved upon for facilitating better grasp of the results.

Response-1: As suggested by the reviewer, we have revised the supplemental figure legends and captions for clarity and improved the layout of Figure S3.

Reviewer 3 Report

The manuscript entitled “A SARS-CoV-2 vaccine designed for manufacturability results in unexpected potency and non-waning humoral response” by Campbell et al demonstrated that MT-001 was considered for manufacturability and ease of distribution, and also Campbell group demonstrate that these attributes are not inconsistent with a highly immunogenic vaccine that confers durable and broad immunity to SARS-CoV-2 and its upcoming new variants. Furthermore, these properties illustrate MT-001 could be a potent new addition to the toolbox of SARS-CoV-2 vaccines and other interventions to prevent infection and limit additional morbidity and mortality from the continuing worldwide pandemic

However, some of the points need to be addressed to enhance the importance of the manuscript:

1. Author need to apply two-way analyses for some of the figures to check the immune responses specifically in the case of doses versus time point of Vaccine administrations such as Figure 3.

2. What is the basis,  author has used a 10 ug dose of MT- 001in SARS cov2 challenged young Hamster  models since in the BL6 mouse models  author has used 1, 3, and 15 ug of MT 001.

3.  It is impressive author has performed long-term studies for MT 001 in Wt, Omicron, and Delta variants, but it could be more significant author could add another group treatment with more any of the potent current vaccines (Pfizer/BioNTech (BNT162b2) and Moderna (mRNA- 1101273). Then it could be compared with the MT 001 administered group.

 4. What justification did the authors sacrifice four days after the SARs CoV2 challenge in hamsters?

5. Please provide a figure representing the lung histology scores difference between PBS, Adjuvant, and MT001 treated groups in Supplementary figure s3.  

6. Did authors observe any physiological changes (Such as body temperature, Body weight, and locomotor activities) in BL6 and Hamster models after the MT 001 vaccine and Sars COv2 administration? 

Author Response

Response to Comments from Reviewer 3:

General comment:  The manuscript entitled “A SARS-CoV-2 vaccine designed for manufacturability results in unexpected potency and non-waning humoral response” by Campbell et al demonstrated that MT-001 was considered for manufacturability and ease of distribution, and also Campbell group demonstrate that these attributes are not inconsistent with a highly immunogenic vaccine that confers durable and broad immunity to SARS-CoV-2 and its upcoming new variants. Furthermore, these properties illustrate MT-001 could be a potent new addition to the toolbox of SARS-CoV-2 vaccines and other interventions to prevent infection and limit additional morbidity and mortality from the continuing worldwide pandemic.

Response: We thank reviewer 3 for their thoughtful review of our manuscript and appreciate their comments on the potential of MT-001.

Comment-1: Author need to apply two-way analyses for some of the figures to check the immune responses specifically in the case of doses versus time point of Vaccine administrations such as Figure 3.

Response-1: We believe that appropriate statistical analyses were performed to test the hypotheses proposed in each experiment presented in this manuscript. In Figure 3, we tested whether adjuvanting with CpG in addition to Alhydrogel resulted in an increased immune response and/or a more balanced Th1/Th2 response as compared to Alhydrogel alone. Thus, a one-tailed t-test is the appropriate statistical method and provides a more stringent test than a two-tailed analysis. In Figure 4, all of the comparisons are pairwise, and thus we used a non-parametric Mann-Whitney test to test for significance between the treated and control groups. Figure 5 looks at statistical differences between groups with multiple independent variables, and thus we used a two-tailed Kruskal-Wallis test.

Comment-2:  What is the basis, author has used a 10 ug dose of MT- 001in SARS cov2 challenged young Hamster models since in the BL6 mouse models author has used 1, 3, and 15 ug of MT 001.

Response-2: In Figure 2, we present a vaccine dose-ranging study in mice, which suggested a 3ug dose was appropriate for subsequent experiments. The 10ug vaccine dose selected for the hamster experiments was based on the 3ug dose used in the mouse experiments, scaled for the difference in body weight, and was consistent with previous literature reports testing similar vaccines in the hamster model used here.

Comment-3: It is impressive author has performed long-term studies for MT 001 in Wt, Omicron, and Delta variants, but it could be more significant author could add another group treatment with more any of the potent current vaccines (Pfizer/BioNTech (BNT162b2) and Moderna (mRNA- 1101273). Then it could be compared with the MT 001 administered group.

Response-3: We thank the reviewer for recognizing the importance of testing the long-term durability of MT-001 and noting its ability to neutralize variants. We agree that the data would have had even more significance had we included a group or groups immunized with the approved mRNA vaccines from Pfizer/BioNTech and Moderna. Unfortunately, this was not feasible as neither Pfizer or Moderna were willing to provide vaccines for these studies (a letter in Nature from Melanie Saville, the Director of Vaccine Development at CEPI, laments the inability to secure approved vaccines for research use: https://www.nature.com/articles/d41586-021-02398-6). Instead, within the discussion section we have compared the titers observed with MT-001 to those reported in the literature for the approved mRNA vaccines in the same model systems and find that in all cases our vaccine performs comparably or better.

Comment-4: What justification did the authors sacrifice four days after the SARs CoV2 challenge in hamsters?

Response-4: In this study, the hamsters were sacrificed at four days post-infection as previous studies have shown viral load in the lungs peaks at this time point which provides the best test of vaccine efficacy (Figure 4) and allows for direct comparison with the aforementioned studies.

Comment-5: Please provide a figure representing the lung histology scores difference between PBS, Adjuvant, and MT001 treated groups in Supplementary figure s3.  

Response-5: We thank the reviewer for bringing this omission to our attention. As suggested by the reviewer, we have revised this figure to include the histology scores from the PBS control group, and also to clarify the figure layout and legend.  In the revised Figure S3, the lung histology scores for PBS, Adjuvant alone and MT-001 appear in the third panel.

Comment-6: Did authors observe any physiological changes (Such as body temperature, Body weight, and locomotor activities) in BL6 and Hamster models after the MT 001 vaccine and Sars COv2 administration? 

Response-6: We would like to clarify that mice were not challenged with SARS-CoV-2 in this work; they were only vaccinated with MT-001. However, in the SARS-CoV-2-infected hamsters, no significant changes in body weight (Figure S3), temperature, or locomotor activities were observed. This information is added in the results section of the revised manuscript.

Round 2

Reviewer 1 Report

The manuscript may be accepted in current form.